# Improved Regret Bounds for Oracle-Based Adversarial Contextual Bandits

**Vasilis Syrgkanis**
Microsoft Research
vasy@microsoft.com

**Haipeng Luo**
Microsoft Research
haipeng@microsoft.com

**Akshay Krishnamurthy**
University of Massachusetts, Amherst
akshay@cs.umass.edu

**Robert E. Schapire**
Microsoft Research
schapire@microsoft.com

## Abstract

We propose a new oracle-based algorithm, BISTRO+, for the adversarial contextual bandit problem, where either contexts are drawn i.i.d. or the sequence of contexts is known a priori, but where the losses are picked adversarially. Our algorithm is computationally efficient, assuming access to an offline optimization oracle, and enjoys a regret of order $O((KT)^{\frac{2}{3}}(\log N)^{\frac{1}{3}})$, where $K$ is the number of actions, $T$ is the number of iterations, and $N$ is the number of baseline policies. Our result is the first to break the $O(T^{\frac{3}{4}})$ barrier achieved by recent algorithms, which was left as a major open problem. Our analysis employs the recent relaxation framework of Rakhlin and Sridharan [7].

## 1 Introduction

We study online decision making problems where a learner chooses an action based on some side information (context) and incurs some cost for that action with a goal of incurring minimal cost over a sequence of rounds. These *contextual online learning* settings form a powerful framework for modeling many important decision-making scenarios with applications ranging from personalized health care to content recommendation and targeted advertising. Many of these applications also involve a partial feedback component, wherein costs for alternative actions are unobserved, and are typically modeled as *contextual bandits*.

The contextual information present in these problems enables learning of a much richer *policy* for choosing actions based on context. In the literature, the typical goal for the learner is to have cumulative cost that is not much higher than the best policy $\pi$ in a large policy class $\Pi$. This is formalized by the notion of *regret*, which is the learner's cumulative cost minus the cumulative cost of the best fixed policy $\pi$ in hindsight.

Naively, one can view the contextual problem as a standard online learning problem where the set of possible "actions" available at each iteration is the set of policies. This perspective is fruitful, as classical algorithms, such as Hedge [5, 3] and Exp4 [2], give information theoretically optimal regret bounds of $O(\sqrt{T\log(N)})$ in full-information and $O(\sqrt{TK\log(N)})$ in the bandit setting, where $T$ is the number of rounds, $K$ is the number of actions, and $N$ is number of policies. However, naively lifting standard online learning algorithms to the contextual setting leads to a running time that is linear in the number of policies. Given that the optimal regret is only logarithmic in $N$ and that our high-level goal is to learn a very rich policy, we want to capture policy classes that are exponentially large. When we use a large policy class, existing algorithms are no longer computationally tractable.

To study this computational question, a number of recent papers have developed *oracle-based algorithms* that only access the policy class through an optimization oracle for the offline full-information problem. Oracle-based approaches harness the research in supervised learning that focuses on designing efficient algorithms for full-information problems and uses it for online and partial-feedback problems. Optimization oracles have been used in designing contextual bandit algorithms [1, 4] that achieve the optimal $O(\sqrt{KT \log(N)})$ regret while also being computationally efficient (i.e. requiring poly$(K, \log(N), T)$ oracle calls and computation). However, these results only apply when the contexts and costs are independently and identically distributed at each iteration, contrasting with the computationally inefficient approaches that can handle adversarial inputs.

Two very recent works provide the first oracle efficient algorithms for the contextual bandit problem in adversarial settings [7, 8]. Rakhlin and Sridharan [7] considers a setting where the contexts are drawn i.i.d. from a known distribution with adversarial costs and they provide an oracle efficient algorithm called BISTRO with $O(T^{\frac{3}{4}} K^{\frac{1}{2}} (\log(N))^{\frac{1}{4}})$ regret. Their algorithm also applies in the transductive setting where the sequence of contexts is known a priori. Srygkanis et. al [8] also obtain a $T^{\frac{3}{4}}$-style bound with a different oracle-efficient algorithm, but in a setting where the learner knows only the set of contexts that will arrive. Both of these results achieve very suboptimal regret bounds, as the dependence on the number of iterations is far from the optimal $O(\sqrt{T})$-bound. A major open question posed by both works is whether the $O(T^{\frac{3}{4}})$ barrier can be broken.

In this paper, we provide an oracle-based contextual bandit algorithm, BISTRO+, that achieves regret $O((KT)^{\frac{2}{3}} (\log(N))^{\frac{1}{3}})$ in both the i.i.d. context and the transductive settings considered by Rakhlin and Sridharan [7]. This bound matches the $T$-dependence of the epoch-greedy algorithm of Langford and Zhang [6] that only applies to the fully stochastic setting. As in Rakhlin and Sridharan [7], our algorithm only requires access to a *value oracle*, which is weaker than the standard argmax oracle, and it makes $K + 1$ oracle calls per iteration. To our knowledge, this is the best regret bound achievable by an oracle-efficient algorithm for any adversarial contextual bandit problem.

Our algorithm and regret bound are based on a novel and improved analysis of the minimax problem that arises in the relaxation-based framework of Rakhlin and Sridharan [7] (hence the name BISTRO+). Our proof requires analyzing the value of a sequential game where the learner chooses a distribution over actions and then the adversary chooses a distribution over costs in some bounded finite domain, with - importantly - a bounded variance. This is unlike the simpler minimax problem analyzed in [7], where the adversary is only constrained by the range of the costs.

Apart from showing that this more structured minimax problem has a small value, we also need to derive an oracle-based strategy for the learner that achieves the improved regret bound. The additional constraints on the game require a much more intricate argument to derive this strategy which is an algorithm for solving a structured two-player minimax game (see Section 4).

## 2  Model and Preliminaries

**Basic notation.**  Throughout the paper we denote with $x_{1:t}$ a sequence of quantities $\{x_1, \dots, x_t\}$ and with $(x, y, z)_{1:t}$ a sequence of tuples $\{(x_1, y_1, z_1), \dots\}$. $\emptyset$ denotes an empty sequence. The vector of ones is denoted by $\mathbf{1}$ and the vector of zeroes is denoted by $\mathbf{0}$. Denote with $[K]$ the set $\{1, \dots, K\}$, $\mathbf{e}_1, \dots, \mathbf{e}_K$ the standard basis vectors in $\mathbb{R}^K$, and $\Delta_U$ the set of distributions over a set $U$. We also use $\Delta_K$ as a shorthand for $\Delta_{[K]}$.

**Contextual online learning.**  We consider the following version of the contextual online learning problem. On each round $t = 1, \dots, T$, the learner observes a context $x_t$ and then chooses a probability distribution $q_t$ over a set of $K$ actions. The adversary then chooses a cost vector $c_t \in [0, 1]^K$. The learner picks an action $\hat{y}_t$ drawn from distribution $q_t$, incurs a cost $c_t(\hat{y}_t)$ and observes only $c_t(\hat{y}_t)$ and not the cost of the other actions.

Throughout the paper we will assume that the context $x_t$ at each iteration $t$ is drawn i.i.d. from a distribution $\mathcal{D}$. This is referred to as the *hybrid i.i.d.-adversarial* setting [7]. As in prior work [7], we assume that the learner can sample contexts from this distribution as needed. It is easy to adapt the arguments in the paper to apply for the transductive setting where the learner knows the sequence of contexts that will arrive. The cost vectors $c_t$ are chosen by a non-adaptive adversary.

The goal of the learner is to compete with a set of policies $\Pi$ of size $N$, where each policy $\pi \in \Pi$ is a function mapping contexts to actions. The cumulative expected regret with respect to the best fixed policy in hindsight is

$$\text{REG} = \sum_{t=1}^{T} \langle q_t, c_t \rangle - \min_{\pi \in \Pi} \sum_{t=1}^{T} c_t(\pi(x_t)).$$

**Optimization value oracle.** We will assume that we are given access to an optimization oracle that when given as input a sequence of contexts and cost vectors $(x, c)_{1:t}$, it outputs the cumulative cost of the best fixed policy, which is

$$\min_{\pi \in \Pi} \sum_{\tau=1}^{t} c_t(\pi(x_t)). \tag{1}$$

This can be viewed as an offline batch optimization or ERM oracle.

## 2.1 Relaxation based algorithms

We briefly review the relaxation based framework proposed in [7]. The reader is directed to [7] for a more extensive exposition. We will also slightly augment the framework with some internal randomness that the algorithm can generate and use, which does not affect the cost of the algorithm.

A crucial concept in the relaxation based framework is the information obtained by the learner at the end of each round $t \in [T]$, which is the following tuple:

$$I_t(x_t, q_t, \hat{y}_t, c_t, S_t) = (x_t, q_t, \hat{y}_t, c_t(\hat{y}_t), S_t),$$

where $\hat{y}_t$ is the realized chosen action drawn from the distribution $q_t$ and $S_t$ is some random string drawn from some distribution that can depend on $q_t$, $\hat{y}_t$ and $c_t(\hat{y}_t)$ and which can be used by the algorithm in subsequent rounds.

**Definition 1** *A partial-information relaxation* $\text{REL}(\cdot)$ *is a function that maps* $(I_1, \ldots, I_t)$ *to a real value for any* $t \in [T]$. *A partial-information relaxation is admissible if for any* $t \in [T]$, *and for all* $I_1, \ldots, I_{t-1}$

$$\mathrm{E}_{x_t} \left[ \min_{q_t} \max_{c_t} \mathrm{E}_{\hat{y}_t \sim q_t, S_t} \left[ c_t(\hat{y}_t) + \text{REL}(I_{1:t-1}, I_t(x_t, q_t, \hat{y}_t, c_t, S_t)) \right] \right] \leq \text{REL}(I_{1:t-1}), \tag{2}$$

*and for all* $x_{1:T}, c_{1:T}$ *and* $q_{1:T}$

$$\mathrm{E}_{\hat{y}_{1:T} \sim q_{1:T}, S_{1:T}} \left[ \text{REL}(I_{1:T}) \right] \geq - \min_{\pi \in \Pi} \sum_{t=1}^{T} c_t(\pi(x_t)). \tag{3}$$

**Definition 2** *Any randomized strategy* $q_{1:T}$ *that certifies inequalities* (2) *and* (3) *is called an admissible strategy.*

A basic lemma proven in [7] is that if one constructs a relaxation and a corresponding admissible strategy, then the expected regret of the admissible strategy is upper bounded by the value of the relaxation at the beginning of time.

**Lemma 1 ([7])** *Let* $\text{REL}$ *be an admissible relaxation and* $q_{1:T}$ *be an admissible strategy. Then for any* $c_{1:T}$, *we have*

$$\mathrm{E}\left[\text{REG}\right] \leq \text{REL}(\emptyset).$$

We will utilize this framework and construct a novel relaxation with an admissible strategy. We will show that the value of the relaxation at the beginning of time is upper bounded by the desired improved regret bound and that the admissible strategy can be efficiently computed assuming access to an optimization value oracle.

## 3   A Faster Contextual Bandit Algorithm

First we define an unbiased estimator the cost vectors $c_t$. In addition to doing the usual importance weighting, we also discretize the estimated cost to either 0 or $L$ for some constant $L \geq K$ to be specified later. Specifically, suppose that at iteration $t$ an action $\hat{y}_t$ is picked based on some distribution $h_t \in \Delta_K$. Now, consider the random variable $X_t$, which is defined conditionally on $\hat{y}_t$ and $h_t$, as

$$X_t = \begin{cases} 1 & \text{with probability } \frac{c_t(\hat{y}_t)}{Lh_t(\hat{y}_t)}, \\ 0 & \text{with the remaining probability.} \end{cases} \tag{4}$$

This is a valid random variable whenever $\min_y h_t(y) \geq \frac{1}{L}$, which will be ensured by the algorithm. This is the only randomness in the random string $S_t$ that we used in the general relaxation framework.

Our construction of an unbiased estimate for each $c_t$ based on the information $I_t$ collected at the end of each round is then: $\hat{c}_t = LX_t \mathbf{e}_{\hat{y}_t}$. Observe that for any $y \in [K]$,

$$\mathrm{E}_{\hat{y}_t, X_t} [\hat{c}_t(y)] = L \cdot \Pr[\hat{y}_t = y] \cdot \Pr[X_t = 1 | \hat{y}_t = y] = L \cdot h_t(y) \cdot \frac{c_t(y)}{Lh_t(y)} = c_t(y).$$

Hence, $\hat{c}_t$ is an unbiased estimate of $c_t$.

We are now ready to define our relaxation. Let $\epsilon_t \in \{-1, 1\}^K$ be a Rademacher random vector (i.e. each coordinate is an independent Rademacher random variable, which is $-1$ or $1$ with equal probability), and let $Z_t \in \{0, L\}$ be a random variable which is $L$ with probability $K/L$ and 0 otherwise. We denote with $\rho_t = (x, \epsilon, Z)_{t+1:T}$ and with $G_t$ the distribution of $\rho_t$ which is described above. Our relaxation is defined as follows:

$$\mathrm{REL}(I_{1:t}) = \mathrm{E}_{\rho_t \sim G_t} [R((x, \hat{c})_{1:t}, \rho_t)], \tag{5}$$

where

$$R((x, \hat{c})_{1:t}, \rho_t) = -\min_{\pi \in \Pi} \left( \sum_{\tau=1}^{t} \hat{c}_\tau(\pi(x_\tau)) + \sum_{\tau=t+1}^{T} 2\epsilon_\tau(\pi(x_\tau))Z_\tau \right) + (T - t)K/L.$$

Note that $\mathrm{REL}(\emptyset)$ is the following quantity, whose first part resembles a Rademacher average:

$$\mathrm{REL}(\emptyset) = 2\mathrm{E}_{(x, \epsilon, Z)_{1:T}} \left[ \max_{\pi \in \Pi} \sum_{\tau=1}^{T} \epsilon_\tau(\pi(x_\tau))Z_\tau \right] + TK/L.$$

Using the following lemma (whose proof is deferred to the supplementary material) and the fact $\mathrm{E}\left[ Z_t^2 \right] \leq KL$, we can upper bound $\mathrm{REL}(\emptyset)$ by $O(\sqrt{TKL \log(N)} + TK/L)$, which after tuning $L$ will give the claimed $O(T^{2/3})$ bound.

**Lemma 2** *Let $\epsilon_t$ be Rademacher random vectors, and $Z_t$ be non-negative real-valued random variables such that $\mathrm{E}\left[ Z_t^2 \right] \leq M$ for some constant $M > 0$. Then*

$$\mathrm{E}_{Z_{1:T}, \epsilon_{1:T}} \left[ \max_{\pi \in \Pi} \sum_{t=1}^{T} \epsilon_t(\pi(x_t)) \cdot Z_t \right] \leq \sqrt{2TM \log(N)}.$$

To show an admissible strategy for our relaxation, we let $D = \{L \cdot \mathbf{e}_i : i \in [K]\} \cup \{\mathbf{0}\}$. For a distribution $p \in \Delta(D)$, we denote with $p(i)$, for $i \in \{0, \ldots, K\}$, the probability assigned to vector $\mathbf{e}_i$, with the convention that $\mathbf{e}_0 = \mathbf{0}$. Also let $\Delta_D' = \{p \in \Delta_D : p(i) \leq 1/L, \forall i \in [K]\}$.

Based on this notation our admissible strategy is defined as

$$q_t = \mathrm{E}_{\rho_t} [q_t(\rho_t)] \quad \text{where} \quad q_t(\rho_t) = \left( 1 - \frac{K}{L} \right) q_t^*(\rho_t) + \frac{1}{L}\mathbf{1}, \tag{6}$$

and

$$q_t^*(\rho_t) = \operatorname*{argmin}_{q \in \Delta_K} \max_{p_t \in \Delta_D'} \mathrm{E}_{\hat{c}_t \sim p_t} [\langle q, \hat{c}_t \rangle + R((x, \hat{c})_{1:t}, \rho_t)]. \tag{7}$$

Algorithm 1 implements this admissible strategy. Note that it suffices to use $q_t(\rho_t)$ for a single random draw $\rho_t$ instead of $q_t$ to ensure the exact same guarantee in expectation. In Section 4 we show that $q_t(\rho_t)$ can be computed efficiently using an optimization value oracle.

We state the main theorem of our relaxation construction and defer the proof to Section 5.

---

**Algorithm 1** BISTRO+

---
**Input:** parameter $L \geq K$
**for** each time step $t \in [T]$ **do**
  Observe $x_t$. Draw $\rho_t = (x, \epsilon, Z)_{t+1:T}$ where each $x_\tau$ is drawn from the distribution of contexts,
  $\epsilon_\tau$ is a Rademacher random vectors and $Z_\tau \in \{0, L\}$ is $L$ with probability $K/L$ and 0 otherwise.
  Compute $q_t(\rho_t)$ based on Eq. (6) (using Algorithm 2).
  Predict $\hat{y}_t \sim q_t(\rho_t)$ and observe $c_t(\hat{y}_t)$.
  Create an estimate $\hat{c}_t = L X_t \mathbf{e}_{\hat{y}_t}$, where $X_t$ is defined in Eq. (4) using $q_t(\rho_t)$ as $h_t$.
**end for**

---

---

**Algorithm 2** Computing $q_t^*(\rho_t)$

---
**Input:** a value optimization oracle, $(x, \hat{c})_{1:t-1}$, $x_t$ and $\rho_t$.
**Output:** $q \in \Delta_K$, a solution to Eq. (7).
Compute $\psi_i$ as in Eq. (9) for all $i = 0, \ldots, K$ using the optimization oracle.
Compute $\phi_i = \frac{\psi_i - \psi_0}{L}$ for all $i \in [K]$.
Let $m = 1$ and $q = \mathbf{0}$.
**for** each coordinate $i \in [K]$ **do**
  Set $q(i) = \min\{(\phi_i)^+, m\}$. $((x)^+ = \max\{x, 0\})$
  Update $m \leftarrow m - q(i)$.
**end for**
Distribute $m$ arbitrarily on the coordinates of $q$ if $m > 0$.

---

**Theorem 3** *The relaxation defined in Equation* (5) *is admissible. An admissible randomized strategy for this relaxation is given by* (6). *The expected regret of* BISTRO+ *is upper bounded by*

$$2\sqrt{2TKL \log(N)} + TK/L, \tag{8}$$

*for any $L \geq K$. Specifically, setting $L = (KT/\log(N))^{\frac{1}{3}}$ when $T \geq K^2 \log(N)$, the regret is of order $O((KT)^{\frac{2}{3}}(\log(N))^{\frac{1}{3}})$.*

## 4   Computational Efficiency

In this section we will argue that if one is given access to a value optimization oracle (1), then one can run BISTRO+ efficiently. Specifically, we will show that the minimizer of Equation (7) can be computed efficiently via Algorithm 2.

**Lemma 4** *Computing the quantity defined in equation* (7) *for any given $\rho_t$ can be done in time $O(K)$ and with only $K + 1$ accesses to a value optimization oracle.*

**Proof:**   For $i \in \{0, \ldots, K\}$, let

$$\psi_i = \min_{\pi \in \Pi} \left( \sum_{\tau=1}^{t-1} \hat{c}_\tau(\pi(x_\tau)) + L\mathbf{e}_i(\pi(x_t)) + \sum_{\tau=t+1}^{T} 2\epsilon_\tau(\pi(x_\tau))Z_t \right) \tag{9}$$

with the convention $\mathbf{e}_0 = \mathbf{0}$. Then observe that we can re-write the definition of $q_t^*(\rho_t)$ as

$$q_t^*(\rho_t) = \operatorname*{argmin}_{q \in \Delta_K} \max_{p_t \in \Delta_D'} \sum_{i=1}^{K} p_t(i)(L \cdot q(i) - \psi_i) - p_t(0) \cdot \psi_0.$$

Observe that each $\psi_i$ can be computed with a single oracle access. Thus we can assume that all $K + 1$ $\psi$'s are computed efficiently and are given. We now argue how to compute the minimizer.

For any $q$, the maximum over $p_t$ can be characterized as follows. With the notation $z_i = L \cdot q(i) - \psi_i$ and $z_0 = -\psi_0$ we re-write the minimax quantity as

$$q_t^*(\rho_t) = \operatorname*{argmin}_{q \in \Delta_K} \max_{p_t \in \Delta_D'} \sum_{i=1}^{K} p_t(i) \cdot z_i + p_t(0) \cdot z_0.$$

Observe that without the constraint that $p_t(i) \leq 1/L$ for $i > 0$ we would put all the probability mass on the maximum of the $z_i$. However, with the constraint the maximizer put as much probability mass as allowed on the maximum coordinate $\operatorname{argmax}_{i \in \{0,\ldots,K\}} z_i$ and continues to the next highest quantity. We repeat this until reaching the quantity $z_0$, which is unconstrained. Thus we can put all the remaining probability mass on this coordinate.

Let $z_{(1)}, z_{(2)}, \ldots, z_{(K)}$ denote the ordered $z_i$ quantities for $i > 0$ (from largest to smallest). Moreover, let $\mu \in [K]$ be the largest index such that $z_{(\mu)} \geq z_0$. By the above reasoning we get that for a given $q$, the maximum over $p_t$ is equal to (recall that we assume $L \geq K$)

$$\sum_{t=1}^{\mu} \frac{z_{(t)}}{L} + \left(1 - \frac{\mu}{L}\right) z_0 = \sum_{t=1}^{\mu} \frac{z_{(t)} - z_0}{L} + z_0.$$

Now since for any $t > \mu$, $z_{(t)} < z_0$, we can write the latter as

$$\sum_{t=1}^{\mu} \frac{z_{(t)}}{L} + \left(1 - \frac{\mu}{L}\right) z_0 = \sum_{i=1}^{K} \frac{(z_i - z_0)^+}{L} + z_0$$

with the convention $(x)^+ = \max\{x, 0\}$. We thus further re-write the minimax expression as

$$q_t^*(\rho_t) = \operatorname*{argmin}_{q \in \Delta_K} \sum_{i=1}^{K} \frac{(z_i - z_0)^+}{L} + z_0 = \operatorname*{argmin}_{q \in \Delta_K} \sum_{i=1}^{K} \frac{(z_i - z_0)^+}{L}$$

$$= \operatorname*{argmin}_{q \in \Delta_K} \sum_{i=1}^{K} \left(q(i) - \frac{\psi_i - \psi_0}{L}\right)^+.$$

Let $\phi_i = \frac{\psi_i - \psi_0}{L}$. The expression becomes: $q_t^*(\rho_t) = \operatorname{argmin}_{q \in \Delta_K} \sum_{t=1}^{K} (q(i) - \phi_i)^+$.

This quantity is minimized as follows: consider any $i \in [K]$ such that $\phi_i \leq 0$. Then putting positive mass $\xi$ on such a coordinate $i$ is going to lead to a marginal increase of $\xi$ in the objective. On the other hand if we put some mass on an index $\phi_i > 0$, then that will not increase the objective until we reach the point where $q(i) = \phi_i$. Thus a minimizer will distribute probability mass of $\min\{\sum_{i:\phi_i > 0} \phi_i, 1\}$, on the coordinates for which $\phi_i > 0$. The remaining mass, if any, can be distributed arbitrarily. See Algorithm 2 for details. ∎

## 5 Proof of Theorem 3

We verify the two conditions for admissibility.

**Final condition.** It is clear that inequality (3) is satisfied since $\hat{c}_t$ are unbiased estimates of $c_t$:

$$\mathrm{E}_{\hat{y}_{1:T}, X_{1:T}} \left[\mathrm{REL}(I_{1:T})\right] = \mathrm{E}_{\hat{y}_{1:T}, X_{1:T}} \left[\max_{\pi \in \Pi} - \sum_{\tau=1}^{T} \hat{c}_\tau(\pi(x_\tau))\right]$$

$$\geq \max_{\pi \in \Pi} - \mathrm{E}_{\hat{y}_{1:T}, X_{1:T}} \left[\sum_{\tau=1}^{T} \hat{c}_\tau(\pi(x_\tau))\right] = \max_{\pi \in \Pi} - \sum_{\tau=1}^{T} c_\tau(\pi(x_\tau)).$$

$t$**-th Step condition.** We now check that inequality (2) is also satisfied at some time step $t \in [T]$. We reason conditionally on the observed context $x_t$ and show that $q_t$ defines an admissible strategy for the relaxation. For convenience let $F_t$ denote the joint distribution of the pair $(\hat{y}_t, X_t)$. Observe that the marginal of $F_t$ on the first coordinate is equal to $q_t$. Let $q_t^* = \mathrm{E}_{\rho_t}[q_t^*(\rho_t)]$. First observe that:

$$\mathrm{E}_{(\hat{y}_t, X_t) \sim F_t} \left[c_t(\hat{y}_t)\right] = \mathrm{E}_{\hat{y}_t \sim q_t} \left[c_t(\hat{y}_t)\right] = \langle q_t, c_t \rangle \leq \langle q_t^*, c_t \rangle + \frac{1}{L} \langle \mathbf{1}, c_t \rangle \leq \mathrm{E}_{(\hat{y}_t, X_t) \sim F_t} \left[\langle q_t^*, \hat{c}_t \rangle\right] + \frac{K}{L}.$$

Hence,

$$\max_{c_t \in [0,1]^K} \mathrm{E}_{(\hat{y}_t, X_t) \sim F_t} \left[c_t(\hat{y}_t) + \mathrm{REL}(I_{1:t})\right] \leq \max_{c_t \in [0,1]^K} \mathrm{E}_{(\hat{y}_t, X_t) \sim F_t} \left[\langle q_t^*, \hat{c}_t \rangle + \mathrm{REL}(I_{1:t})\right] + \frac{K}{L}.$$

We now work with the first term of the right hand side.

$$\max_{c_t \in [0,1]^K} \mathrm{E}_{(\hat{y}_t, X_t) \sim F_t} \left[ \langle q_t^*, \hat{c}_t \rangle + \mathrm{REL}(I_{1:t}) \right]$$

$$= \max_{c_t \in [0,1]^K} \mathrm{E}_{(\hat{y}_t, X_t) \sim F_t} \left[ \mathrm{E}_{\rho_t \sim G_t} \left[ \langle q_t^*(\rho_t), \hat{c}_t \rangle + R((x, \hat{c})_{1:t}, \rho_t) \right] \right]$$

Observe that $\hat{c}_t$ is a random variable taking values in $D$ and such that the probability that it is equal to $L\mathbf{e}_y$ (for $y \in \{0, \dots, K\}$) can be upper bounded as

$$\Pr[\hat{c}_t = L\mathbf{e}_y] = \mathrm{E}_{\rho_t \sim G_t} \left[ \Pr[\hat{c}_t = L\mathbf{e}_y | \rho_t] \right] = \mathrm{E}_{\rho_t \sim G_t} \left[ q_t(\rho_t)(y) \frac{c_t(y)}{L \cdot q_t(\rho_t)(y)} \right] \leq 1/L.$$

Thus we can upper bound the latter quantity by the supremum over all distributions in $\Delta_D'$, i.e.,

$$\max_{c_t \in [0,1]^K} \mathrm{E}_{(\hat{y}_t, X_t)} \left[ \langle q_t^*, \hat{c}_t \rangle + \mathrm{REL}(I_{1:t}) \right] \leq \max_{p_t \in \Delta_D'} \mathrm{E}_{\hat{c}_t \sim p_t} \left[ \mathrm{E}_{\rho_t \sim G_t} \left[ \langle q_t^*(\rho_t), \hat{c}_t \rangle + R((x, \hat{c})_{1:t}, \rho_t) \right] \right].$$

Now we can continue by pushing the expectation over $\rho_t$ outside of the supremum, i.e.,

$$\max_{c_t \in [0,1]^K} \mathrm{E}_{(\hat{y}_t, X_t)} \left[ \langle q_t^*, \hat{c}_t \rangle + \mathrm{REL}(I_{1:t}) \right] \leq \mathrm{E}_{\rho_t \sim G_t} \left[ \max_{p_t \in \Delta_D'} \mathrm{E}_{\hat{c}_t \sim p_t} \left[ \langle q_t^*(\rho_t), \hat{c}_t \rangle + R((x, \hat{c})_{1:t}, \rho_t) \right] \right]$$

and working conditionally on $\rho_t$. Since the expression is linear in $p_t$, the supremum is realized, and by the definition of $q_t^*(\rho_t)$, the quantity inside the expectation $\mathrm{E}_{\rho_t \sim G_t}$ is equal to

$$\min_{q \in \Delta_K} \max_{p_t \in \Delta_D'} \mathrm{E}_{\hat{c}_t \sim p_t} \left[ \langle q, \hat{c}_t \rangle + R((x, \hat{c})_{1:t}, \rho_t) \right].$$

We can now apply the minimax theorem and upper bound the above by

$$\max_{p_t \in \Delta_D'} \min_{q \in \Delta_K} \mathrm{E}_{\hat{c}_t \sim p_t} \left[ \langle q, \hat{c}_t \rangle + R((x, \hat{c})_{1:t}, \rho_t) \right].$$

Since the inner objective is linear in $q$, we continue with

$$\max_{p_t \in \Delta_D'} \min_{y} \mathrm{E}_{\hat{c}_t \sim p_t} \left[ \hat{c}_t(y) + R((x, \hat{c})_{1:t}, \rho_t) \right].$$

We can now expand the definition of $R(\cdot)$

$$\max_{p_t \in \Delta_D'} \min_{y} \mathrm{E}_{\hat{c}_t \sim p_t} \left[ \hat{c}_t(y) + \max_{\pi \in \Pi} - \left( \sum_{\tau=1}^{t} \hat{c}_\tau(\pi(x_\tau)) + \sum_{\tau=t+1}^{T} 2\epsilon_\tau(\pi(x_\tau)) Z_\tau \right) \right] + (T-t)K/L.$$

With the notation $A_\pi = - \sum_{\tau=1}^{t-1} \hat{c}_\tau(\pi(x_\tau)) - \sum_{\tau=t+1}^{T} 2\epsilon_\tau(\pi(x_\tau)) Z_\tau$, we re-write the above as

$$\max_{p_t \in \Delta_D'} \min_{y} \mathrm{E}_{\hat{c}_t \sim p_t} \left[ \hat{c}_t(y) + \max_{\pi \in \Pi}(A_\pi - \hat{c}_t(\pi(x_t))) \right] + (T-t)K/L.$$

We now upper bound the first term. The extra term $(T-t)K/L$ will be combined with the extra $K/L$ that we have abandoned to give the correct term $(T-(t-1))K/L$ needed for $\mathrm{REL}(I_{1:t-1})$.

Observe that we can re-write the first term by using symmetrization as

$$\max_{p_t \in \Delta_D'} \min_{y} \mathrm{E}_{\hat{c}_t \sim p_t} \left[ \hat{c}_t(y) + \max_{\pi \in \Pi}(A_\pi - \hat{c}_t(\pi(x_t))) \right]$$

$$= \max_{p_t \in \Delta_D'} \mathrm{E}_{\hat{c}_t \sim p_t} \left[ \max_{\pi \in \Pi}(A_\pi + \min_{y} \mathrm{E}_{\hat{c}_t' \sim p_t} [\hat{c}_t'(y)] - \hat{c}_t(\pi(x_t))) \right]$$

$$\leq \max_{p_t \in \Delta_D'} \mathrm{E}_{\hat{c}_t \sim p_t} \left[ \max_{\pi \in \Pi}(A_\pi + \mathrm{E}_{\hat{c}_t' \sim p_t} [\hat{c}_t'(\pi(x_t))] - \hat{c}_t(\pi(x_t))) \right]$$

$$\leq \max_{p_t \in \Delta_D'} \mathrm{E}_{\hat{c}_t, \hat{c}_t' \sim p_t} \left[ \max_{\pi \in \Pi}(A_\pi + \hat{c}_t'(\pi(x_t)) - \hat{c}_t(\pi(x_t))) \right]$$

$$= \max_{p_t \in \Delta_D'} \mathrm{E}_{\hat{c}_t, \hat{c}_t' \sim p_t, \delta} \left[ \max_{\pi \in \Pi}(A_\pi + \delta \left( \hat{c}_t'(\pi(x_t)) - \hat{c}_t(\pi(x_t)) \right)) \right]$$

$$\leq \max_{p_t \in \Delta_D'} \mathrm{E}_{\hat{c}_t \sim p_t, \delta} \left[ \max_{\pi \in \Pi}(A_\pi + 2\delta \hat{c}_t(\pi(x_t))) \right]$$

where $\delta$ is a random variable which is $-1$ and $1$ with equal probability. The last inequality follows by splitting the maximum into two equal parts.

Conditioning on $\hat{c}_t$, consider the random variable $M_t$ which is $-\max_y \hat{c}_t(y)$ or $\max_y \hat{c}_t(y)$ on the coordinates where $\hat{c}_t$ is equal to zero and equal to $\hat{c}_t$ on the coordinate that achieves the maximum. This is clearly an unbiased estimate of $\hat{c}_t$. Thus we can upper bound the last quantity by

$$\max_{p_t \in \Delta'_D} \mathrm{E}_{\hat{c}_t \sim p_t, \delta} \left[ \max_{\pi \in \Pi} (A_\pi + 2\delta \mathrm{E}\left[M_t(\pi(x_t))|\hat{c}_t\right]) \right] \leq \max_{p_t \in \Delta'_D} \mathrm{E}_{\hat{c}_t, \delta, M_t} \left[ \max_{\pi \in \Pi} (A_\pi + 2\delta M_t(\pi(x_t))) \right].$$

The random vector $\delta M_t$, conditioning on $\hat{c}_t$, is equal to $-\max_y \hat{c}_t(y)$ or $\max_y \hat{c}_t(y)$ with equal probability independently on each coordinate. Moreover, observe that for any distribution $p_t \in \Delta'_D$, the distribution of the maximum coordinate of $\hat{c}_t$ has support on $\{0, L\}$ and is equal to $L$ with probability at most $K/L$. Since the objective only depends on the distribution of the maximum coordinate of $\hat{c}_t$, we can continue the upper bound with a maximum over any distribution of random vectors whose coordinates are $0$ with probability at least $1 - K/L$ and otherwise are $-L$ or $L$ with equal probability. Specifically, let $\epsilon_t$ be a Rademacher random vector, we continue with

$$\max_{Z_t \in \Delta_{\{0,L\}}: Pr[Z_t = L] \leq K/L} \mathrm{E}_{\epsilon_t, Z_t} \left[ \max_{\pi \in \Pi} (A_\pi + 2\epsilon_t(\pi(x_t)) Z_t) \right].$$

Now observe that if we denote with $a = \Pr[Z_t = L]$, the above is equal to

$$\max_{a : 0 \leq a \leq K/L} \left( (1-a) \max_{\pi \in \Pi} (A_\pi) + a \mathrm{E}_{\epsilon_t} \left[ \max_{\pi \in \Pi} (A_\pi + 2\epsilon_t(\pi(x_t)) L) \right] \right).$$

We now argue that this maximum is achieved by setting $a = K/L$. For that it suffices to show that

$$\max_{\pi \in \Pi} (A_\pi) \leq \mathrm{E}_{\epsilon_t} \left[ \max_{\pi \in \Pi} (A_\pi + 2\epsilon_t(\pi(x_t)) L) \right],$$

which is true by observing that with $\pi^* = \mathrm{argmax}_{\pi \in \Pi}(A_\pi)$ one has

$$\mathrm{E}_{\epsilon_t} \left[ \max_{\pi \in \Pi} (A_\pi + 2\epsilon_t(\pi(x_t)) L) \right] \geq \mathrm{E}_{\epsilon_t} \left[ A_{\pi^*} + 2\epsilon_t(\pi^*(x_t)) L) \right] = A_{\pi^*} + \mathrm{E}_{\epsilon_t} \left[ 2\epsilon_t(\pi^*(x_t)) L) \right] = A_{\pi^*}.$$

Thus we can upper bound the quantity we want by

$$\mathrm{E}_{\epsilon_t, Z_t} \left[ \max_{\pi \in \Pi} (A_\pi + 2\epsilon_t(\pi(x_t)) Z_t \right],$$

where $\epsilon_t$ is a Rademacher random vector and $Z_t$ is now a random variable which is equal to $L$ with probability $K/L$ and is equal to $0$ with the remaining probability.

Taking expectation over $\rho_t$ and $x_t$ and adding the $(T - (t-1))K/L$ term that we abandoned, we arrive at the desired upper bound of $\mathrm{REL}(I_{1:t-1})$. This concludes the proof of admissibility.

**Regret bound.** By applying Lemma 2 (See Appendix A) with $E[Z_t^2] = L^2 \Pr[Z_t = L] = KL$ and invoking Lemma 1, we get the regret bound in Equation (8). ∎

## 6 Discussion

In this paper, we present a new oracle-based algorithm for adversarial contextual bandits and we prove that it achieves $O((KT)^{2/3} \log(N)^{1/3})$ regret in the settings studied by Rakhlin and Sridharan [7]. This is the best regret bound that we are aware of among oracle-based algorithms.

While our bound improves on the $O(T^{3/4})$ bounds in prior work [7, 8], achieving the optimal $O(\sqrt{TK \log(N)})$ regret bound with an oracle based approach still remains an important open question. Another interesting avenue for future work involves removing the stochastic assumption on the contexts.

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
