[Supplementary Material]

# A    A Supplementary Lemma

**Lemma 2.**  *Let $\epsilon_t$ be Rademacher random vectors, and $Z_t$ be non-negative real-valued random variables, such that $\mathrm{E}\left[Z_t^2\right] \leq M$. Then:*

$$\mathrm{E}_{Z_{1:T},\epsilon_{1:T}}\left[\max_{\pi\in\Pi}\sum_{t=1}^{T}\epsilon_t(\pi(x_t))\cdot Z_t\right] \leq \sqrt{2TM\log(N)}$$

**Proof:**

$$
\begin{aligned}
\mathrm{E}_{Z_{1:T},\epsilon_{1:T}}\left[\max_{\pi\in\Pi}\sum_{t=1}^{T}\epsilon_t(\pi(x_t))\cdot Z_t\right] &= \mathrm{E}_{Z_{1:T}}\left[\frac{1}{\lambda}\mathrm{E}_{\epsilon_{1:T}}\left[\log\left(\max_{\pi\in\Pi}e^{\lambda\sum_{t=1}^{T}\epsilon_t(\pi(x_t))\cdot Z_t}\right)\right]\right] \\
&\leq \mathrm{E}_{Z_{1:T}}\left[\frac{1}{\lambda}\log\left(\mathrm{E}_{\epsilon_{1:T}}\left[\max_{\pi\in\Pi}e^{\lambda\sum_{t=1}^{T}\epsilon_t(\pi(x_t))\cdot Z_t}\right]\right)\right] \\
&\leq \mathrm{E}_{Z_{1:T}}\left[\frac{1}{\lambda}\log\left(\mathrm{E}_{\epsilon_{1:T}}\left[\sum_{\pi\in\Pi}e^{\lambda\sum_{t=1}^{T}\epsilon_t(\pi(x_t))\cdot Z_t}\right]\right)\right] \\
&= \mathrm{E}_{Z_{1:T}}\left[\frac{1}{\lambda}\log\left(\sum_{\pi\in\Pi}\mathrm{E}_{\epsilon_{1:T}}\left[\prod_{t=1}^{T}e^{\lambda\epsilon_t(\pi(x_t))\cdot Z_t}\right]\right)\right] \\
&= \mathrm{E}_{Z_{1:T}}\left[\frac{1}{\lambda}\log\left(\sum_{\pi\in\Pi}\prod_{t=1}^{T}\mathrm{E}_{\epsilon_t}\left[e^{\lambda\epsilon_t(\pi(x_t))\cdot Z_t}\right]\right)\right].
\end{aligned}
$$

Now observe that $\mathrm{E}_{\epsilon_t}\left[e^{\lambda\epsilon_t(\pi(x_t))\cdot Z_t}\right] = \frac{e^{\lambda\cdot Z_t}+e^{-\lambda\cdot Z_t}}{2} \leq e^{\lambda^2\cdot Z_t^2/2}$. Thus

$$
\begin{aligned}
\mathrm{E}_{Z_{1:T},\epsilon_{1:T}}\left[\max_{\pi\in\Pi}\sum_{t=1}^{T}\epsilon_t(\pi(x_t))\cdot Z_t\right] &\leq \mathrm{E}_{Z_{1:T}}\left[\frac{1}{\lambda}\log\left(\sum_{\pi\in\Pi}\prod_{t=1}^{T}e^{\lambda^2\cdot Z_t^2/2}\right)\right] \\
&= \mathrm{E}_{Z_{1:T}}\left[\frac{1}{\lambda}\log\left(Ne^{\lambda^2\sum_{t=1}^{T}Z_t^2/2}\right)\right] \\
&= \frac{1}{\lambda}\log(N) + \lambda\mathrm{E}_{Z_{1:T}}\left[\sum_{t=1}^{T}Z_t^2/2\right] \\
&\leq \frac{1}{\lambda}\log(N) + \lambda MT/2.
\end{aligned}
$$

Optimizing over $\lambda$ yields the result. ∎