[Reviews · NeurIPS 2016]

Reviewer 1

Summary

The paper describes an oracle-based algorithm (for use in the adversarial contextual bandit problem) with better time complexity than existing algorithms under the same set of assumptions.

Qualitative Assessment

The paper addresses the problem of determining policies for use in an adversarial contextual bandit problem, and describes an algorithm which improves the previously published best bound on regret. Both the explanation and analysis of the algorithm are difficult to follow, and could be vastly improved by presentation of proper pseudo-code style algorithms - there is a need to define all inputs and outputs, as well as detailing the computational steps (rather than the vague descriptions in the paper). It is not clear how much of the computational effort is simply shifted to the oracle in this approach, and a simple illustrative example would be a great help in understanding this work properly.

Confidence in this Review

2-Confident (read it all; understood it all reasonably well)


Reviewer 2

Summary

The paper gives the first efficient algorithm for the hybrid i.i.d.-adversarial contextual bandit problem that has a \tilde{O}(T^{2/3}) regret bound. The previous best result was \tilde{O}(T^{3/4}). The paper builds heavily on the relaxation technique of Rakhlin and Sridharan.

Qualitative Assessment

The paper is an important contribution to the literature on contextual bandit algorithms.

Confidence in this Review

1-Less confident (might not have understood significant parts)


Reviewer 3

Summary

In this paper, the authors study a particular setting of contextual bandit problem. The time horizon is known. The contexts are assumed to be iid. At each step, a distribution of bounded loss with bounded variance is chosen by an adversary when the context is drawn. The learner has an unlimited access to the distribution of contexts, and an access to an Oracle which outputs the value of the cumulative loss of the best policy for any sequence of (context,loss). In comparison to the hybrid iid-adversarial setting proposed recently in "Bistro: An efficient Relaxation-Based Method for Contextual Bandits", the difference is that the losses are not deterministic. In this setting the player chooses a distribution over actions, and then the adversary chooses a distribution over losses. The authors use the same relaxation framework proposed for the contextual bandit problem in the above reference. Their main result (Theorem 3) shows that in the case of the proposed setting the relaxation algorithm reaches a regret in O(T^2/3) versus O(T^3/4) for the above reference.

Qualitative Assessment

This work is incremental in comparison to the above reference. Although the main result shows an improvement of the regret bound, almost the same setting, the same relaxation algorithm and the same maths are used. So the step seems to be sub-standard for NIPS. This paper seems to be written in a hurry. The idea behind the approach is not exposed. The algorithm is not explained. The paper is hard to follow. There are a lot of notations, which are not usefull. For instance the random string S_t introduced with the relaxation approach (lines 96-97) seems to be X_t (lines 120-121). So why S_t and X_t ? Some notations are used, and defined in a later paragraph. For instance e_i is used line 123 and defined line 137. Some variables are never defined (what is N?) or implicitly defined (M). Some notations are used only once: R_\pi = REL(0), so why R_\pi is used lines 130-142 ? All of this acts as an obstacle to understanting. I read the answers of authors. I move my rating about novelty.

Confidence in this Review

2-Confident (read it all; understood it all reasonably well)


Reviewer 4

Summary

The authors studies the contextual MAB problem. The features across the periods are generated i.i.d. from a distribution known to the decision maker, but the losses across the periods are adversarially generated. By designing a new "partial information relaxation" and using a recent framework for contextual MAB proposed by Rakhlin and Sridharan, the authors manage to improve the regret from the state of the art, namely O(T^(3/4)), to O(T^(2/3)). The key to improvement is the incorporation of a Rademacher term (the middle term) in the partial information relaxation defined between Line 129 and 130. This term has two Rademacher random variable, Z_t and epsilon_t. My understanding is that the rv Z_t ensures that the partial information relaxation is valid (satisfying eqns (2, 3)), while the rv epsilon_t ensures that the relaxation term grows in a rate of sqrt(T) (conf Lemma 2) instead of T, which brings about the improvement.

Qualitative Assessment

The paper makes some interesting contribution by proposal a new partial information relaxation to improve the regret bound. The analysis of the partial info relaxation bears some similarity to that in Rahklin and Sridharan, but the inclusion of the Radamacher term is new. By using Lemma 2, the bound is improved. While this is definitely a new result and there are new ideas, the similarity to current literature kind of compels me to give a '3' instead of a '4' for the Novelty/Originality and Technical Constribution scores. Presentation: The paper is concise and well written, but I think that the proof of admissability between 5 and 7 is too "straight-lined". The unraveling and bounding from Line 156 to Line 188 is rather long, but it would be more readable if the authors first highlight the major steps in the calculation, and then explain how to go from one major step to the next. This will give a better feel on your design of partial information relaxation. In particular, I think you should label "the quantity inside the expectation" between Line 162 and 163 to avoid ambiguity. A clear presentation of the proof is very important, since it will guide future readers on the design of partial information relaxation in the framework.

Confidence in this Review

2-Confident (read it all; understood it all reasonably well)


Reviewer 5

Summary

This paper gives a efficient algorithm in Oracle-Based Adversarial Contextual Bandits problem which improves the regret bound of this problem.

Qualitative Assessment

I do not really understand the contents in this paper. But I think that the idea is good, and the algorithm designed in this paper indeed improve the regret bound of this problem.

Confidence in this Review

1-Less confident (might not have understood significant parts)


Reviewer 6

Summary

The paper introduces an oracle-based adversarial contextual bandit problem under transductive assumption. Authors prove a formal regret bound of order O((KT)^{1/3}(\log N)^{1/3}), which improves previous work with O(T^{3/4}) regret bound.

Qualitative Assessment

The paper is based on a novel relaxation based approach proposed by Rakhlin and Sridharan [7]. Authors achieve an improved regret bound O((KT)^{1/3}(\log N)^{1/3}), comparing with O(T^{3/4}) regret bound in previous work. The sketch of the proof is clear and seems solid (however I did not check all proofs in detail). The result is sound and very promising, and is deserved to be published. Typos: line 20 "not much higher then the best policy" -> "not much higher than the best policy"

Confidence in this Review

1-Less confident (might not have understood significant parts)